# Chyawanprash: A Traditional Indian Bioactive Health Supplement

**DOI:** 10.3390/biom9050161

**Published:** 2019-04-26

**Authors:** Rohit Sharma, Natália Martins, Kamil Kuca, Ashun Chaudhary, Atul Kabra, Meda M. Rao, Pradeep Kumar Prajapati

**Affiliations:** 1Central Ayurveda Research Institute for Drug Development (CCRAS), Ministry of AYUSH, Government of India, 4-CN Block, Sector-V, Bidhannagar, Kolkata-700091, India; medamrao69@gmail.com; 2Faculty of Medicine, University of Porto, Alameda Prof. Hernani Monteiro, 4200-319 Porto, Portugal; 3Institute for research and Innovation in Heath (i3S), University of Porto, Rua Alfredo Allen, 4200-135 Porto, Portugal; 4Department of Chemistry, Faculty of Science, University of Hradec Králové, Rokitanskeho 62, 50003 Hradec Králové, Czech Republic; kamil.kuca@uhk.cz; 5Department of Biotechnology, MMEC, Maharishi Markandeshwar (Deemed to be University), Mullana, Ambala-133207, India; ashun.chaudhary@gmail.com; 6Department of Pharmacology, Kota College of Pharmacy, Kota, Rajasthan-324005, India; atul.kbr@gmail.com; 7Department of Rasashastra & Bhaishajya Kalpana, All India Institute of Ayurveda, New Delhi-110076, India; prajapati.pradeep1@gmail.com

**Keywords:** Chyawanprash, ayurveda, Indian gooseberry, medicinal plants, nutraceutical, health supplement, traditional medicine

## Abstract

Chyawanprash (CP) is an Ayurvedic health supplement which is made up of a super-concentrated blend of nutrient-rich herbs and minerals. It is meant to restore drained reserves of life force (*ojas*) and to preserve strength, stamina, and vitality, while stalling the course of aging. Chyawanprash is formulated by processing around 50 medicinal herbs and their extracts, including the prime ingredient, *Amla* (Indian gooseberry), which is the world’s richest source of vitamin C. Chyawanprash preparation involves preparing a decoction of herbs, followed by dried extract preparation, subsequent mixture with honey, and addition of aromatic herb powders (namely clove, cardamom, and cinnamon) as standard. The finished product has a fruit jam-like consistency, and a sweet, sour, and spicy flavor. Scientific exploration of CP is warranted to understand its therapeutic efficacy. Scattered information exploring the therapeutic potential of CP is available, and there is a need to assemble it. Thus, an effort was made to compile the scattered information from ancient Ayurvedic texts and treatises, along with ethnobotanical, ethnopharmacological, and scientifically validated literature, that highlight the role of CP in therapeutics. Citations relevant to the topic were screened.

## 1. Introduction

Chyawanprash (CP) (also known as chyavanaprasha, chyavanaprash, chyavanaprasam, and chyawanaprash) comprises two lexes, “*Chyawan*” and “*Prasha*”. The word *Chyawan* is the name of a sage, and also symbolizes ‘degenerative change’. *Prasha* denotes a drug or foodstuff that is suitable for consumption. Indeed, CP is a comprehensive ‘metabolic’ tonic; it contains a variety of herbs and is used to promote health and prevent diseases. Chyawanprash is an ancient Indian formulation (a polyherbal jam), prepared according to a traditional Ayurvedic recipe, enriched with several herbs, herbal extracts, and processed minerals. Regarded by many experts as an essential health supplement, CP has been around for centuries. Chyawanprash possesses multiple health benefits and has been widely used since ancient times as a health supplement and as a medicine for enhancing immunity and longevity. Chyawanprash has been a part of every Indian’s life from the day it was introduced, irrespective of sociocultural, political, and scientific factors. It was one of the most appreciated foods for its antiaging effects long before vitamins, minerals, and antioxidant supplements came into existence [1].

*Rasayana,* a branch of Ayurveda, includes a number of specialized approaches aimed at prolonging life, preventing aging and diseases, eliminating degenerative processes, and promoting excellent health. Of all the *Rasayana* formulations enumerated during the classical and medieval periods, CP undoubtedly stands out as the most important. This formulation has made major strides as an over-the-counter product since it entered the consumer market in the 1950s. It is highly appreciated for possessing multiple health benefits and addressing the preventive, promotive, and curative aspects of health.

Chyawanprash consists of *Amla*/*Amalaki* (*Phyllanthus emblica*/Indian gooseberry) pulp as a base, and this is considered to be the most effective *Rasayana* for sustaining homeostasis [2]. Chyawanprash that contains *Amla* has a mixed taste, combining sweet, sour, bitter, pungent, and astringent qualities [3]. On regular intake, it maintains physiological functions and rejuvenates the whole body system [4].

## 2. Origin

The atypical name of CP originates from the legend of *Chyawan Rishi*, who was a forest sage. Various ancient sacred treatises, such as the *Mahabharata*, the *Puranas*, etc., describe how the twin *Ashwini Kumar* brothers (the royal physicians to the Gods during the Vedic era) invented this polyherbal preparation to make the sage *Chyawan Rishi* younger and improve his vitality and strength. The formulation was prepared at his hermitage at Dhosi Hill near the Narnaul area, in the state of Haryana, India; drawing its name from the ancient sage, the formula was called “Chyawanprash”. He followed strict practices to become enlightened, and this had made him weak, emaciated, and aged. To regain his youthfulness, vitality, and strength, he used CP [5,6,7,8]. The foremost historically recorded recipe for CP is reported in the *Charaka Samhita*, the ancient Ayurvedic classic, where it is appreciated as being superior to all other herbal rejuvenative tonics [9].

## 3. Composition

Chyawanprash is a potent antioxidant paste, prepared through the synergistic blending of around 50 herbs and spices. Chyawanprash falls, by virtue of its consistency and form of dosage, under the category of *Awaleha* (electuaries/herbal jams), a group of Ayurvedic formulations [10]. Typically, CP includes four classes of herbal drugs: The *Dashmula* class (ten roots); the *Chaturjata* class (four aromatic plants); *Ashtavarga* (threatened medicinal herbs from the Northwest Himalayas, which are not commercially available in the modern era) [11]; and a general class (materials not belonging to the former classes). The Chyawanprash formula is described in the ancient Ayurvedic texts, namely, Ashtanga Hridayam, Charaka Samhita, Sangandhara Samhita, which are dedicated to clinical management. The dominant ingredient is *Amla*, a citrus fruit that is a highly renowned and potent botanical in Ayurveda. The main ingredients of CP, along with their botanical identities, key active biomolecules and specific therapeutic roles, are detailed in Figure 1 and Table 1 [12,13,14,15,16,17,18,19,20,21,22,23,24,25,26,27,28,29,30,31,32,33,34,35,36,37,38,39,40,41,42,43,44,45,46,47,48,49,50,51,52,53,54,55].

Each ingredient of CP is scientifically validated for its nutritive and therapeutic efficacy. All these nutrients are blended in specific quantities and subjected to unique pharmaceutical processes in such a fashion that builds a potent synergy for optimal health virtues [56,57]. However, noncompliance with ancient manufacturing guidelines and deviation from the original recipe is a common malpractice in the pharma sector. The original formula is wiped out in between the practices of preparation and immoral marketing tactics to make it more presentable and appealing than the competitors’ product. It will no longer be called CP if there is any change in the ingredients of the traditional formula [58].

## 4. The “Missing” 8 Ingredients of Chyawanprash

The Government of India has already framed policies for the preservation, cultivation, and sustainable extraction of rare and endangered medicinal plants. Out of these, eight rare herbs that are ingredients of the “original” ancient recipe for CP are missing in commercial formulations of this traditional medicine. Commercial formulations now use substitute herbs. Among these eight, four belong to the orchid family, three are from the lily family, and one belongs to the house of gingers. These are conjointly called *Ashtavarga* and are said to augment the antioxidant role of *Amla*. Non-availability of authentic plants, confusion in vernacular names, and lack of chemical markers lead to substitution/adulteration of *Ashtavarga* plants [59]. These *Ashtavarga* herbs which are close to extinction are stipulated in Table 2.

## 5. Manufacturing Process of Chyawanprash

Since the standard operating procedure (SOP) for CP preparation is not clearly narrated in ancient literature, at present, the modus operandi differs for each manufacturer. Current authoritative books of the Indian system of medicine mention the use of 500 numbers of *Amla* in a single lot. However, variation in *Amla* size and the quantity of the obtained pulp is the major limiting factor for the SOP and standardization. In historical times, *Amla* was mainly collected from forests. It was observed that *Amla* fruit collected from forests has more concentration of vitamin C than that coming from cultivated fruit [60]. The weight of 500 fresh *Amla* varies from 2.5–25.25 kg, as cultivated hybrid *Amla* has a bigger size than wild collected *Amla*. The Ayurvedic Formulary of India (AFI) considers the use of 2.5 kg of fresh *Amla* for 500 numbers and 2.4 kg sugar for its preparation [10]. If cultivated hybrid *Amla* is available, 500 such fruits would weigh approximately 6.5 kg. 

The standard method of preparation of CP is: 50 g of each of medicinal herbs, such as *Bael, Agnimanth, Kashmarya, Shyonak, Paatla, Gokshur, Sarivan, Barikateri, Kantakaari, Kakdasingi, Draaksha, Haritaki, Guduchi, Bala, Bhumyamalaki, Vasa, Jivanti, Kachur, Pushkarmul, Musta, Mudagparni, Mashaparni, Shalparni, Pithawan, Pipali, Kaknasa, Varahi, Vidaarikand, Punarnava, Neelkamal, Aguru, Chandan, Shatavar*, and *Asgandh*, are suspended in 16 L potable water. Five-hundred *Amla* fruits (each fruit having a weight of around 15–20 g, total weight: 6.5 kg) are swathed in clean cotton cloth to form a bale (*pottali*) and submerged into the aforementioned combination of herbs. Thereafter, the admixture is boiled until decoction is reduced to 1/4th. After taking off the *pottali*, seeds are removed from *Amla*; the remaining pulpy portion is rubbed on a clean muslin cloth, *Amla* fibers are separated, and *Amlapishthi* (wet paste of *Amla* pulp) is collected. Decoction is then strained, and mare is discarded. After this, *Amlapishthi* is mixed with *Yamakadravyas* (lipids: 500 g cow ghee and sesame oil each) in an iron container and fried until it gets brownish-red and the *Yamaka* (lipids) starts separating. Sugar syrup is then prepared by adding sugar in the herbal decoction. Fried *Amlapishthi* is added to this decoction syrup and heated until attainment of viscidity of two strings. Then, when the heating is stopped, *Prakshepadravya* (herbal powders of 150 g *Vanshalochan*; 100 g *Pipali* and *Nagakesar*; Elaichi, *Tamalpatra* and *Dalchini*, 10 g each) are added and stirred until a homogeneous mixture is obtained. After cooling the mixture, 250 g honey (old, natural, pure) is uniformly mixed, and the finished product is obtained and packed in airtight sterile containers. Finally, the prepared CP is of a dark brown color, having wet paste-like appearance and consistency. The whole unit operating process of traditional CP preparation is depicted in Figure 2.

Boiling of *Amla* fruits with decoction and the subsequent cooking processes might be inducing pH change, release of acid soluble contents, hydrolysis/cleavage of various bioactive molecules, extraction of soluble chemicals, exchange with intra/extra cellular chemicals of mixture, and several suitable phytochemical interactions to make an ideal blend of this nutraceutical. Some Ayurvedic additives, *Shukti Bhasma* (pearl oyster calx) 100 g, *Abhraka Bhasma* (mica calx) 100 g, *Shringa Bhasma* (deer horn calx) 100 g, *Makardhawaja* (preparation of red sulphide of mercury and gold) 25 g, clove 25 g and *Rajata* (silver foil) 75 in number, for special health benefits, are also added by some manufacturers [1].

## 6. Mode of Administration

Chyawanprash can be used by all age groups in every season, as its ingredients nullify the unpleasant effects of intense weather and climate or environmental change [2,61]. Chyawanprash should be taken in a quantity such that it does not interfere with hunger and appetite for food [62,63,64,65,66]. The general dosage of CP (12–28 g) is to be taken with milk (100–250 ml) on an empty stomach in the morning [66,67].

However, it is advocated that individuals suffering from asthma/respiratory ailments should avoid intake of milk and curd [67]. In such cases, the formulation can be administered with lukewarm water. It is recommended to consume CP within a year from the manufacturing date, as a study has indicated that chemical deterioration may occur during the storage period, resulting in loss of the therapeutic potency of CP [68].

## 7. Phytochemical and Quality Specifications of Chyawanprash

Chyawanprash is a semi-solid sticky paste with a brownish black appearance, chiefly having sweet and spicy odor, with a sweet and astringent feel after taste with aroma of *Prakshepadravya* (powder of seven herbs) [69,70]. The taste is predominantly governed by the flavors of honey, cow ghee (clarified butter), and *Triphala* (a mixture of three myrobalans), and the aroma by cow ghee and certain spices viz. sandalwood, cinnamon, and cardamom. Limited studies are available on quality testing of CP. A major part in the composition of CP is *Amla*, which is rich in vitamin C and polyphenolics, including flavonoids. The phenolic compounds of CP possess antioxidant principles that are said to contribute to the rejuvenating and tonic attributes of CP. A high-performance liquid chromatography (HPLC) analysis has identified several biologically active phenolics in CP, i.e., gallic acid, protocatechuic acid, catechin, caffeic acid, vanillic acid, chlorogenic acid, syringic acid, rutin, ferulic acid, and quercitrin, which may account for its therapeutic activity [71]. By contrast, individual pharmaceutical companies have their own in-house specifications for the quality of CP, which are not in the public domain. The Ayurvedic Pharmacopoeia of India (API) has published a monograph on CP along with a brief method of preparation and various physicochemical and assay tests as official quality standards. These include description, identification (such as microscopy, thin layer chromatography (TLC), physicochemical parameters (loss drying, total ash, acid-insoluble ash, alcohol-soluble extractive, water-soluble extractive, pH)), assay, microbial limit, and test for aflatoxin. The Ayurvedic Pharmacopoeia of India mentions that CP should contain no less than 0.5% of gallic acid when assayed, based on the officially stated method [70].

## 8. The ‘Vitamin C’ Controversy 

*Amla*, having rich vitamin C (445 mg/100 g) contents, constitutes the main ingredient (35%) [72,73]. Owing to the lack of uniform quality control standards of Ayurvedic drugs, it becomes challenging to ensure the uniformity of their composition and so the efficacy of final products [74]. Although the official quality testing methods for CP [75] do not contain vitamin C content, there are contrasting findings apropos of its presence in CP [76,77], possibly due to the application of less sensitive and nonspecific methods of investigation. A study in 1997 found that vitamin C was missing in the tested CP samples, and it might have been destroyed during cooking of the *Amla* pulp with cow ghee in the pharmaceutical process [78]. It has been reported that upon heat exposure during preparation of CP, the vitamin C contents remain unaffected [68,79], with a study reporting 34 mg/100 g vitamin C in CP [80]. Another study found that the percentage of vitamin C in the old samples of CP (0.0253 ± 0.0001%) was much lower than that of the new samples (0.0512 ± 0.0003%), thus signifying the chances of degradation on storage [68].

## 9. Chyawanprash: A Nutraceutical and Functional Food

The term ‘nutraceutical’ was coined in 1989 by Stephen De Felice as “a food or part of a food that provides medical or health benefits, including the prevention and/or treatment of disease.” Chyawanprash has been a consistent part of Indian tradition both as a functional food and nutraceutical for the past 5000 years, with constant zeal and vivacity, and has survived owing to its peerless health benefits. Chyawanprash is reported to have rich vitamin, protein, dietary fiber, energy contents, carbohydrate, low fat contents (no-*trans* and zero percent cholesterol), and appreciable levels of major and minor trace elements (mg/100g), such as Fe (21.1), Zn (3.1), Co (3.7), Cu (0.667), Ni (1.4), Pb (2.4), Mn (8.3), vitamin C (0.5), tannic acid (20.2), other vitamins A, E, B1, B2, and carotenoids that act as micronutrients for health-invigorating purposes. It also provides several essential phytoconstituents, namely, flavonoids, alkaloids, saponins, antioxidants, piperine, phenolic compounds, etc. The synergistic antioxidant effects of vitamin C along with vitamin E and carotenoids are well known. The rich nutritive composition and antioxidant biomolecules of CP act both singly as well as synergistically for immuno-modulation, body building, health restoration, and prevention of oxidative damage (a leading cause of several degenerative diseases) [81,82,83].

## 10. Health Benefits

### 10.1. Ancient Claims and Contemporary Scientific Evidence

Traditional Ayurveda practitioners call CP an “Ageless Wonder”. The formula of CP is time-tested and is still effective to mitigate the present world’s health concerns. In the context of CP, Charaka Samhita narrates: ‘It is the premier Rasayana, beneficial for allaying cough, asthma and other respiratory ailments; it nourishes the weak and degenerating tissues, promotes vigour, vitality and is anti-ageing’. As per ancient classics, regular intake of this tonic helps to attain intellect, memory, immunity, freedom from disease, endurance, improved functioning of the senses, great sexual strength and stamina, improved digestive processes, improvised skin-tone and glow, and restores/maintains the normal biofunctions of *Vata* (bodily humor regulating all movements, circulations and neuroconductive actions) [3,84].

Chyawanprash helps to balance the three *doshas*—*Vata*, *Pitta*, and *Kapha* (bodily humors/bioenergies regulating the structure and biofunctions of the human body). In the Ayurvedic perspective, the specific actions of herbs in CP in the micro and macronutrient supplement level, metabolic level, and tissue nourishment level are well recognized [85]. Chyawanprash has passed the scrutiny of several scientific studies. Contemporary studies corroborate and validate the ancient claims and traditional beliefs regarding its therapeutic use. The herbal and spicy ingredients of CP help to convalesce the circulatory system, thus channelizing the removal of the toxins from distant tissues and visceral organs. It builds a congruent synergy amid physiological functions steering toward an improved metabolism. All herbal and natural products in the composition of CP have been well investigated and explored by the scientific community for their therapeutic vistas. It is very challenging to uncover the active phytochemicals, the rationality behind its therapeutic usage, and the underlying mechanistic role of herbal medicine by adopting contemporary scientific tools and methods. However, this does not imply that all the doctrines or beliefs in traditional medical systems which are not justifiable by scientific substantiation are irrational and non-existent. It is aptly cited in Charaka Samhita, “What is perceptible to humans is merely a petite fraction of this cosmos and what we cannot observe is far more than that, which doesn’t make that non-existent”. Chyawanprash is beneficial for health in several ways. It is an excellent ergogenic (enhancing physical performance), tonic, rejuvenator, anabolic, immunomodulator and promotes strength to the gastrointestinal tract, digestive organs, cardiovascular, respiratory, and cerebrospinal systems, neuronal circuits, and renal and reproductive tissues [86].

### 10.2. Improves Digestion and Metabolism

Chyawanprash helps to eliminate the accumulated excreta via improving digestion and excretion. It is reported to alleviate nausea, vomiting, hyperacidity, dyspepsia, and flatulence. Chyawanprash has also been found to relieve gastritis, peptic ulcer, gut cramps, and correct the gastrointestinal functions. It purifies blood, works as detoxifier, and promotes healthy liver function [1,64]. It protects and strengthens the liver and kidneys and improves lipid and protein metabolism [87,88,89,90,91]. The herbs of CP, such as *Nagakesar, Tejpatra, Ela, Dalchini. Paatla, Agnimanth, Gambhari, Bael, Shyonak. Sarivan. Draaksha, Haritaki,* honey*, Bhumyamalaki, Kachur, Pushkarmul, Musta, Kaknasa. Vidaarikand*, and *Aguru*, help to improve digestion and metabolism [92,93].

It is common practice to add the nourishing honey and cow ghee (clarified butter) in certain Ayurvedic herbal formulations to act as “a transporter of potency of herbs” (aka *Yogavahi* in Ayurveda), and it is believed to promote the quick absorption and assimilation of various herbal constituents in the distant tissues (lacto-vegan diet comprising milk and milk products is strongly recommended in Ayurveda). In the case of CP, its sweet flavor favors its quick assimilation and facilitates better passage of its active ingredients into cell walls [94,95].

### 10.3. Protect and Strengthens the Respiratory System

A regular intake of CP strengthens the trachea–bronchial tree and hence improves the immunity and functioning of the respiratory system. It helps to treat respiratory infections, allergic cough, asthma, bronchospasm, rhinitis, seasonal or nonseason respiratory disorders, common cold, and tuberculosis, and thus strengthens the respiratory system. It is also used as an adjunct to antitubercular drugs to augment their bioactivity and prevent their side effects [96,97,98]. *Pipali, Kantakaari, Kakdasingi, Bhumyamalaki, Vasa, Pushkarmul, Prishnaparni, Agnimanth, Shalparni,* sesame oil, and *Amla* help to nourish the respiratory system [92,93,99]. In a randomized controlled trial (RCT), 90 pulmonary tuberculosis patients were treated with CP 10 g, twice daily as an adjunct to antitubercular drugs. CP augmented the bioactivity of antitubercular drugs and prevented their side effects. Cough, expectoration, weakness, loss of appetite, loss of weight, fever, edema aches, and hemoptysis disappeared almost completely in the treated group, along with improvement in the hemoglobin (Hb) levels and effective healing as evidenced through chest X-ray post-therapy [96,100]. Another observational study on 99 newly diagnosed pulmonary tuberculosis patients revealed that concomitant adjunct use of CP with antitubercular drugs significantly abated the symptoms and improved bioavailability of isoniazid and pyrazinamide [98].

### 10.4. Antioxidant, Adaptogenic, and Immune-Booster

The combination or cocktail of phytocompounds (as in CP) offers better antioxidant effects than single antioxidant therapy [101]. The adaptogenic characteristics of CP are attributable to its excellent antiaging and anxiolytic supplement. The revitalizing and tonic effects of CP could be due to its rich antioxidant composition, bioactive phytoconstituents, such as carotenoids, flavonoids, tannins, and phenolic compounds [102,103,104,105], though supportive experimental and clinical evidence is scarce. Recent investigations have ascertained that polyphenols (gallic acid, catechin, epicatechin) in CP exert key antioxidant potential and is known to possess potent neuroprotective, cytoprotective, and antioxidant properties [106,107]. Piperine content in CP act as a bioavailability enhancer [107]. Chyawanprash is an effective adaptogenic [108]. Some clinical reports do support the adaptogenic and antioxidant effect of CP on normal and depressive subjects [109].

A study evaluated and ascertained highly potent free radical scavenging, based on the synthetic DPPH (1,1-diphenyl,2-picrylhydrazyl) scavenging and antioxidant activities of ethyl acetate extracts of several market brands of CP. The findings were proximate to the standard ascorbic acid (IC_50_ 20.69 µg/mL) [110]. Another study found potent DPPH radical scavenging ability and antioxidant effects of ethanolic extracts of CP [107]. Chyawanprash strengthens immunity and facilitates the healing process [111]. Due to the rich *Amla* percentage, CP is loaded in high vitamin C, polyphenolics, including flavonoids, and exhibits evident antioxidant and free radical scavenging activity, enhances the immune system, and fights infections [112]. Vitamin C also helps to revive and restore the energy loss of the human body [113]. Vitamin C conjugates to gallic acid molecules and reducing sugars and facilitates the development of intricate synergistic effects with other phytoconstituents [114]. Polyphenols are acknowledged to be more effective antioxidants in vitro than vitamin E and C on a molar basis. Polyphenolic compounds in several herbs and natural honey in CP are found beneficial in various human degenerative diseases, cardiovascular disorders, and diabetes [83]. Several natural antioxidants, especially flavonoids, exert multiple bioactivities, including antibacterial, antiviral, anti-inflammatory, antiallergic, antithrombotic, and vasodilator effects [115]. 

In a 6-month-long randomized, open labelled, prospective, multicenter, clinical study in children (5–12 years), CP was shown to lead to significant improvement in immunity, energy levels, physical strength, vigor, and quality of life assessed through KIDSCREEN QOL-27 questionnaires in children [116]. 

An experimental study showed that CP pretreatment significantly reduced plasma histamine levels and serum immunoglobulin E (IgE) release when rats and mice were challenged with allergen- and ovalbumin-induced allergy, respectively. This suggests the antiallergic potential of CP. Natural killer (NK) cell activity was significantly (versus dimethyl sulfoxide) increased in different concentration ratios of NK cells and target cells by CP treatment. On treating dendritic cells with CP, a significant increase in the secretions of tumor necrosis factor-alpha (TNF-α) and macrophage inflammatory protein-1 alpha (MIP-1α), stimulation in interleukin-1 beta (IL-1β) levels, and rise in phagocytic activity were observed. The augmented immunity marker levels (TNF-α, IL-1β, and MIP1α), as well as enhancement of NK cells and phagocytic activity support the immunomodulatory properties of CP [117]. Clinical studies also support the immune-booster role of CP as demonstrated by reduced disease symptoms of seasonal influences, modulated IgE and immunity markers C3 and C4 levels, improved pulmonary functions, decreased cortisol levels, and increased quality of life (QoL) [118].

The minute quantities of spice components of CP are also known for their wide range of health benefits by their antioxidative, chemopreventive, antimutagenic, anti-inflammatory, immune-modulatory effects on cells and several beneficial effects on the gastrointestinal, cardiovascular, respiratory, metabolic, reproductive, neural, and other systems [83]. 

### 10.5. Nootropic Potential 

CP nourishes the brain cells, harmonizes neuronal activities, improves memory, and enhances learning ability, storage, recall, and intellect. It relaxes the central nervous system (CNS), thereby acting as an anxiolytic and an antidepressive, and alleviates insomnia. Research has also suggested its procholenergic activity and antiamnesic potential [119,120,121]. The rich *Amla* and ascorbic acid contents play a vital role in such activities [122,123]. *Musta, Vidaarikand, Neelkamal, Aguru, Nagakesar, Guduchi, Ashwagandha, Shalparni, Prishnaparni*, and *Amla* possess potent antioxidant and anti-inflammatory properties, thereby improving CNS functions [92,93]. In a double-blind, placebo-controlled trial on 60 participants (normal volunteers and patients suffering from depression), CP achieved a significantly effective reduction in the Hamilton-D (HAM-D) scores compared to placebo in both normal subjects and patients of depression [69]. In an RCT on 128 college students, CP significantly improved cognitive functions, i.e., alertness, attention, and concentration [120]. 

### 10.6. Cardiotonic Value

Chyawanprash is a potent cardiotonic. It strengthens the structure and functions of the heart and corrects the heart pumping rhythm by recuperating blood flow to its musculature. Chyawanprash is also reported to correct blood disorders and improve structure and functions of the vascular system. Chyawanprash also exerts antihyperlipidemic activity and alleviates metabolic impairments [80,124]. Components of CP—*Amla, Neelkamal, Punarnawa, Pushkarmul, Kachur, Vasa, Bala, Sarivan, Pithawan, Barikateri* and *Gokshur*—are well-recognized in their ability to rejuvenate and restore the cardiovascular system functions [92,93]. *Amla* has shown antiatherogenic, anticoagulant, hypolipidemic, antihypertensive, antioxidant, antiplatelet, and vasodilatory effects, as well as lipid deposition inhibitory properties [125]. In rat models, *Punarnawa* increased the reduced level of glutathione (GSH), superoxide dismutase (SOD) catalase (CAT) and decreased the elevated level of malondialdehyde (MDA) in cardiac tissue [126].

### 10.7. Potent Aphrodisiac and Balances the Endocrine System

Regular intake of CP improves sexual life, boosts virility, and fertility in each gender. It improves the functioning of gonads, strengthens the endocrine system, and balances the hormonal flow. It improves semen quality in males and the menstrual cycle in females [127,128,129]. Ingredients such as *Gokshur, Varahikand,* sesame oil*, Shatavari, Vidaarikand, Bala, Jivanti, Mudagparni, Mashaparni, Ashwagandha*, and *Vanshalochan* contribute to the aphrodisiac and vitalizing properties of CP [92,93]. A recent systematic review on *Ashwagandha* demonstrated statistical (*p* ≤ 0.002 versus baseline) increase in sperm concentration (167%), semen volume (59%), sperm motility (57%), serum testosterone (17%) and luteinizing hormone (34%) levels, in oligospermic males after 90 days of *Withania somnifera* treatment [130]. In rat models, *Gokshur* and *Shatavari* also showed significant improvement in sexual behavior, androgen levels, increased amount and intromission frequency, improved penile erection, testosterone level, and sperm count [131].

### 10.8. Radioprotective, Cytoprotective, Genoprotective, Antimutagenic and Anticarcinogenic Effects 

A study evaluating the radioprotective effect of CP in mice subjected to a lethal dose of gamma-radiation revealed that CP could provide good radioprotection at a minimal nontoxic dose. The best radioprotection was observed for 15 mg/kg, where the higher number of survivors was found after completion of 30 days post-irradiation [132]. A study found genoprotective efficacy of CP (Dabur company) on the somatic chromosomes of 25 bidi smokers (bidi: A traditional handmade conical smoking stick, prepared by filling tobacco in *Diospyros melanaxylon* leaves). A total of 20 g of CP was administered for two months, twice a day, and parameters such as the mitotic index (MI), chromosomal aberrations (CA), sister chromatid exchanges (SCE), and satellite associations (SA) were studied and found to have significantly decreased (*p* < 0.01) in CP-fed smokers compared to normal smokers. A significant decline in the frequency of CA is indicative of the genoprotective role of CP against mutagenic agents present in tobacco smoke [109].

A cytogenetic study established the genoprotective and antioxidant potential of CP in oral premalignant cancer (conducted on 21 betel quid chewing oral pre-cancerous lesions subjects) [133]. Due to its rich *Amla* contents, CP also exhibits cytoprotective (effective against metal clastogens), anticarcinogenic, and antimutagenic activities and stabilizes the side-effects of chemotherapy and radiotherapy [134,135,136,137,138,139]. The synergistic effects of the cocktail of herbal metabolites of CP and the multiple points of intervention offer higher efficacy during chemoprevention regimens [140]. In an RCT on 75 patients of head and neck cancer, CP (10 g, twice daily) along with radiotherapy reduced the severity of mucosal reactions and improved the Hb levels [69].

### 10.9. Favorable Effects on Lipid Profile and Glycaemic Levels

Owing to its rich sugar and honey contents, CP is generally considered to be contraindicated in diabetics; however, contrary to this widespread belief, CP is reported to reduce postprandial hyperglycemia in the oral glucose tolerance test and substantially reduce blood cholesterol level compared to vitamin C [80]. Chyawanprash is also an efficient hypolipidemic [141]. A study conducted on CP for evaluation of health promotion in elderly people reported a decrease in cholesterol, triglycerides, LDL (low-density lipoprotein), and increase in HDL (high-density lipoprotein) levels which corroborates its indications in geriatrics as cited in *Phalashruti* (beneficial effects) of this formulation [142]. In a randomized open label clinical study (*n* = 121; age group: 18–70) on type 2 diabetics, no statistically significant change in HbA1c and blood sugar levels was found. This signifies the safety of CP in type 2 diabetic patients controlled by oral hypoglycemic agents. Additionally, a statistically significant convalescence was also found in energy levels of diabetics [143].

### 10.10. Other Preventive, Promotive and Curative Health Benefits

Chyawanprash helps in better absorption of calcium and protein synthesis, thereby strengthening bones and teeth, and improving muscle tone. It also promotes growth in juveniles and helps in gaining weight. Its profound Rasayana effect due to potent herbs like *Amla*, *Guduchi*, and *Ashwagandha* helps to balance the body’s natural processes and modulate the neuroendocrine-immune activities [144,145,146]. It eliminates blood impurities and acts as a natural detox [147]. It promotes hair growth, skin complexion, cures dermal infections, and improvises personality characteristics by imparting splendor, exquisiteness, youthfulness, wisdom, vitality, and glow [92,93]. In hairless mice model, CP has shown a protective effect on photoaging of skin. In HeLa cells, CP suppressed epidermal thickening, improved the proliferation of human keratinocytes, and effectively removed ROS (reactive oxygen species), which are liable for skin photoaging [148]. In a study, CP showed promising potential for use as an antimicrobial agent. CHCl_3_ as well as hydrolyzed CHCl_3_ extract of CP showed concentration-dependent antimicrobial activity [149]. Chyawanprash has also shown protective effects in steroid-induced opacities in the eye lens of a chick embryo [150]. In a double-blind, placebo-controlled study on 177 subjects, CP improved Hb levels consistently, irrespective of the season of its consumption, along with improvement in pulmonary function tests and immunological parameters [69].

## 11. Toxicity and Safety Concerns

Although numerous works have been carried out on this formulation, no evident information on toxicity has been available until now. If taken in prescribed dosage, CP is considered to be safe. A report suggested that *Amla* should not be consumed at bedtime to avert ill effects on teeth [151]. Chyawanprash is rich in *Amla* contents; therefore, it is better to avoid it at bedtime. Usually, this information is missing from CP packaging.

## 12. Necessity for Standards of Chyawanprash

Based on the available market samples, it was found that the consistency and flavor of CP vary from company to company. These differences are even noted within the same pharma company in different manufacturing batches. Market surveys have shown great variation in the ingredients/composition of CP in many brands. In addition, texture and appearance may vary from a smooth to a slightly grainy consistency, color may vary from lighter brown to dark and shiny brown, and taste has also been found to be sweet, sour or spicy. Real classical CP is sour and less sweet, while current pharma sector generally adds high sugar levels and make it sweeter to make it more palatable. Even batch variations apropos of CP preparation are also observed in pharma companies; hence, there is immense need for maintaining high raw materials quality and finished product standards. 

Chyawanprash covers a large area of the market, as it is being endorsed as a health supplement or nutraceutical product by its manufacturers, not as Ayurvedic medicine. It is the need of the hour for policy makers, the Indian Pharmacopoeia, the Ministry of Health and Family Welfare, and the Bureau of Indian Standards (functioning under Ministry of Consumer Affairs) to take the responsibly and work in this area to formulate CP’s uniform standards.

## 13. Market Trends

Indian regulations permit the manufacture and sale of CP either as an Ayurvedic medicine by following exactly the recipe and the process as per the authoritative text listed in the regulations (referred to commonly as classical Ayurvedic medicine) or as a proprietary Ayurvedic medicine, where modification to an authoritative text-based recipe is allowed as long as all the ingredients are listed in any one of the texts officially recognized by the law [152]. 

Despite several negative points related to commercialization of this traditional formulation of CP, it is still a widely accepted and used health/nutrition supplement among Indian consumers. It is being advised and used by all age groups and in several health conditions. In the Indian market, CP is certainly distinguished from other nutraceuticals based on its visibility on market shelves, brand variants, huge number of ads, and a cultural acceptability. The market value of CP in 2010 was over 4 billion (about $80 million USD), which makes CP India’s best-selling Ayurvedic medicine [153]. There are many CP brands in the Indian market, such as Dabur, Emami Group, Himalaya, Bajaj, and Baidyanath; however, the leading brand is Dabur, with a market share of 70%. Comparative test performance scores of various leading market brands of CP are detailed in Table 3 [154].

Chyawanprash is losing its real meaning and efficacy because of an upsurge in immoral market trends and noncompliance with ancient manufacturing guidelines. Only the name remains the same, but the ingredients and the preparation totally vary from company to company. Thus, there is a need to get a hold in the market, as the original efficacy of CP is being compromised.

Companies are launching CP in cookies, sugar-free biscuits, snack bars, chocolate granules, fruit-flavored (orange or mango) variants as part of a bid to make the ‘traditional’ brand appealing to young consumers [155,156,157,158]. Apart from introducing new and exciting variants of CP, organizations have backed their marketing campaigns with well-known stars. To entice the younger generation, Indian movie stars and sportsman such as Akshay Kumar, Shahrukh Khan, Ravi Kishan, Virat Kohli, Saina Nehwal, Sachin Tendulkar, and M.S. Dhoni have been featured in CP advertisements and other promotional activities [153]. The cost of production of CP, if it is produced in bulk, will be around ₹ 70–80 per kg. However, there are no stringent regulations on companies for pricing, as price control systems are not applicable to Ayurveda products.

## 14. Perspectives and Future Directions

Chyawanprash is a traditional recipe which is manufactured and has been popularized by firms. Each firm has held information generated by them as propriety and has not published them in research journals, perhaps to maintain ownership of this information and link to their product. This could also limit contemporary scientists’ ability to access and review data on traditional products through online searches that they are used to [159]. 

This traditional product is a complex mixture comprising dozens of active phytocompounds with very broad biological effects on different targets. In such a complex nature of product, it is very challenging to describe in detail the efficacy supported by the mechanism of actions. Moreover, in this form, the synergistic or antagonistic effects of compounds are difficult to deal with and are not evincible from the present literature. Nevertheless, a wide scope is open for future researchers to reach better conclusions. 

Though information from available in vitro and in vivo studies is still limited, the following recommendations are warranted to validate multitherapeutic claims of CP: (1) Extensive, well-stratified, multicenter RCTs of longer durations with a larger sample size and longer follow-up; (2) clinical evaluation of short-term and long-term effects of CP supplementation; (3) comparative in vitro and in vivo investigations on classical and marketed products of CP; (4) evaluating the benefits accruing with the adjunct use of CP with other therapeutic agents on different targets; (5) and studies identifying biochemical and molecular targets of CP.

## 15. Conclusions

Natural health products with medicinal value are gaining importance in clinical research as they offer better alternatives, owing to fewer side-effects and cost-effectiveness than conventional synthetic nutraceuticals. Among the vast library of such products, CP is immensely valuable in terms of therapeutics and global trade. This review underscores the plethora of ancient therapeutic claims of CP, coupled with their validation by available scientific evidence. Reported evidence supports its multifaceted preventive, promotive, and curative health benefits; proving it to be an ancient elixir with a modern cure. However, mechanistic studies and sufficient clinical reports are still lacking. Despite the traditional implementation in ayurvedic medicine and the reported efficacy evidence, there is a requirement of controlled experiments on the effect of the main active compounds and their synergistic or antagonistic effect in order to clarify their mechanism of action. This could also lead to improvement of the available market brands of CP that is not necessarily the optimal version. In fact, strict compliance with the centuries-old recipe alone is not per se a guarantee of success in absence of appropriate scientific evidence. Nevertheless, the present report can be used for future investigations as well as clinical purposes. To sum up, CP is an Ayurvedic superfood and healer par excellence that strengthens the immune system and revitalizes the psychosomatic system, a superior, nutritious, and safe health tonic that is beneficial for all age groups and genders alike. 

## Figures and Tables

**Figure 1 biomolecules-09-00161-f001:**
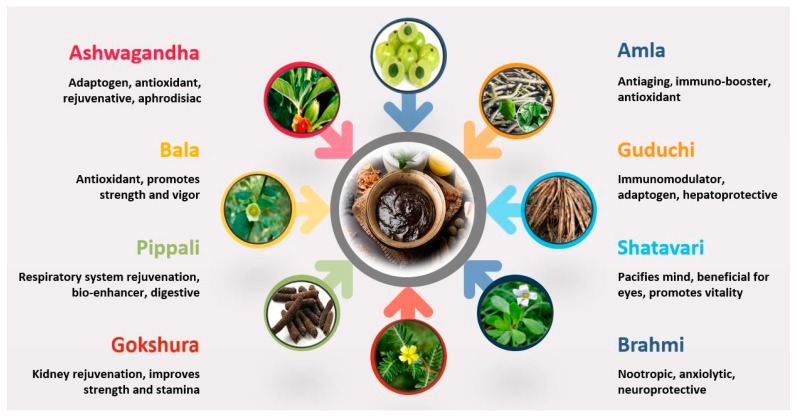
Key plant ingredients of Chyawanprash and their health benefits.

**Figure 2 biomolecules-09-00161-f002:**
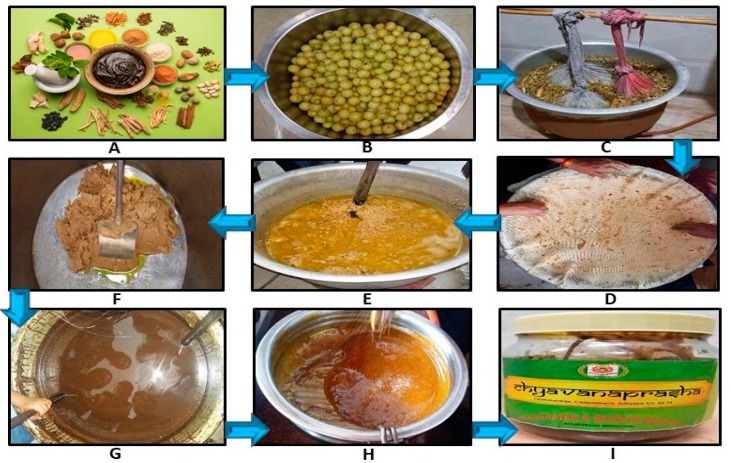
Unit operating process of traditional Chyawnprash preparation. (**A**) Raw material collection; (**B**) fresh *Amla* taken; (**C**) *Amla* boiling in *pottali* suspended in herbal decoction; (**D**) *Amla* pulp separated in muslin cloth; (**E**) *Amla* pulp frying in ghee and sesame oil; (**F**) fried until pulp goes brownish-red and the lipids start separating; (**G**) *Pishthi* cooked in decoction syrup until the attainment of two strings viscidity; (**H**) upon cooling, *prakshepa* herbal powders and honey added and mixed homogeneously; (**I**) finally, prepared Chyawanprash packed in airtight sterile containers.

**Table 1 biomolecules-09-00161-t001:** Ingredients of CP with their botanical identities and specific therapeutic roles.

Medicinal Plants/Special Additives	Major Active Biomolecule	Common Name	Therapeutic Role
*Adhatoda vasica* Nees	Vasicine, Vasicine acetate, aasicinone, aasicoline, 2-acetyl benzyl amine and adhatodine	*Vasaka, Vasa, Arusa*	Antiulcer, bronchodilator, expectorant, anti-allergic, cholagogue, cardiovascular and respiratory disorders
*Aegle marmelos* Correa	Gallic acid, quercetin, and rutin	*Bael, Bel*	Antidiarrheal, gastroprotective, anti-ulcerative, radioprotective
*Aquilaria agallocha* Roxb.	Kusunol, jjinkohol, 10-epi-γ-eudesmol, vanillic acid, aquilarone derivatives and phenylethyl chromones	*Aguru, Akil*	Antiasthmatic, anti-inflammatory, analgesic, carminative, antimicrobial
*Bambusa arundinacea* Willd	Oxalic acid, resins, waxes, benzoic acid, taxiphyllin, diferuloyl arabinoxylanhexasaccharide and oligosaccharide	*Vanshalochana*	Stimulant, astringent, antiulcer, emmenagogue, aphrodisiac
*Boerhavia diffusa* Linn.	Flavonoids, alkaloids, glycosides, rotenoids, steroids, and triterpenoids	*Punarnawa, Gadahapuran, Gadahbindo*	Antiaging, antioxidant, antipyretic, diuretic, anti-inflammatory, hematinic
Cinnamomum tamala Nees and Ebrrn.	Essential oil, camphor, linalool, p-cymene, o-cymene, and 1,8-cineole, Jeolikote, (E)-cinnamaldehyde, (E)-cinnamyl acetate, (E)-innamaldehyde, 1,8-cineol, and eugenol	Tejpat, Tejpatra	Stimulant, antiulcer, antimicrobial, antidepressant, anticancer, digestive, hepatoprotective
*Cinnamomum zeylanicum* Breyn	α-bergamotene, α -copaene, α - humulene, δ-cadinene, tetradecanol and viridiflorene, Linalool, eugenol, β –caryophyllene, (E)-cinnamaldehyde, α -terpineol, (E)-cinnamyl acetate, (E)- caryophyllene, caryophyllene oxide, α –cadinol, tetradecanal and globulol	*Dalchini*	Hematinic, gastroprotective, antinociceptive, digestive, appetizer, hepatoprotective
*Curcuma zedoaria* Rose.	8,9-dehydro9-formyl-cycloisolongifolene, 6-ethenyl-4,5,6,7-tetrahydro-3,6-dimethyl-5-isopropenyl-trans-benzofuran, eucalyptol and γ-elemene	*Kachur, Ban haldi, narkachur*	Antidiarrheal, tonic, stimulant, useful in flatulence and dyspepsia
*Cyperus rotundus* Linn.	Amentoflavone, ginkgetin, isoginkgetin, sciadopitysin, Cyperene, Humulen and Selinene, Zierone, Campholenic Aldehyde, Pinene, Longiverbenone, Vatirenene, Copaene, Limonene, Terpineol, Azulene, Selinene, Myrtenol, Calacorene, Fokienol, Isogermacrene D, and Isolongifolene	*Nagarmotha, Mustak, Musta*	Stimulant, hepatoprotective, stomachic, diuretic, antispasmodic, carminative, emmenagogue, anti-inflammatory, antirheumatic
*Desmodium gangetium* DC Pennel	N-dimenthyltryltryptamine, Hypaphorine, hordenine, caudicine, Gangetin-3H, Gangetinin and desmodin	*Shalparni, Sarivan*	General debility, fatigue, neuroprotective, cardiovascular and respiratory disorders
*Elettaria cardamomum* Maton	1,8-cineole, α-terpinyl acetate, sabinene, 4-terpinen-4-ol, and myrcene	*Elaichi, Cardamom*	Stimulant, stomachic, tonic, appetizer, useful in nausea, heartburn and intestinal spasms
*Emblica officinalis* Gaertn.	Galic acid, ellagic acid, 1-O galloyl-beta-D-glucose, 3,6-di-O-galloyl- D-glucose, chebulinic acid, quercetin, chebulagic acid, corilagin, 1,6- di-O -galloyl beta-D-glucose, 3-Ethylgallic acid, and isostrictiniin	*Amalaki, Amla,* Indian Gooseberry	Immunomodulatory, rejuvenative, neuroprotective, hepatoprotective, antioxidant, cardiotonic, enhance general vitality, cognition and promotes longevity
*Gmelina arborea* Roxb.	(Z)-3-hexenol, 1-octen-3-ol, hexanol, heptacosane, pentacosane, 1-pentacosene, nonanal and (E)-2-decenal	*Gambhari, Gamhar, Kashmarya*	Promotes virility, strength and lactation
*Inula racemose* Hook.	Eudesmanolide, elemanolide, germacranolide, sesquicaranolide, guainolide, humulane, and heptadeca-1,8,11,14-tetraene	*Pushkarmul*	Antihistaminic, bronchodilator, cures cough, cold, asthma and flank pain.
*Leptadenia reticulata* Wight and Am.	α & β amyrin, ferulic acid, luteolin, diosmetin, rutin, β-sitosterol, stigmasterol, hentriacontanol, simiarenol, apigenin, reticulin, deniculatin, leptaculatin, lupanol 3-O diglucoside, leptidine 1, luteolin, and diosmtin	*Jivanti*	Stimulant, restorative, nutrient, anticancer, aphrodisiac, improves vision, immunity and life expectancy
*Martynia diandra* Glox.	Pelargonidin-3-5-diglucoside, cyanidin-3- galactoside, p-hydroxy benzoic acid, gentisic acid, arachidic acid, linoleic acid, palmitic acid, stearic acid, apigenin, apigenin-7-o-glucuronide	*Ulatkanta*	Hepatotonic, cholagogue, laxative, anorexia, indigestion, constipation
*Mesua ferrea* Linn.	Mesuaferrin A, mesuaferrin B, mesuaferrin C, caloxanthone C, macluraxanthone, 1,5-dihydroxyxanthone and tovopyrifolin C	*Nagakesar*	Antitoxic, cardiotonic, carminative, digestive, relieves urinary tract disorders, gout and swelling
*Nelumbium speciosum Willd.*	Liensinine, isoliensinine, neferine, nuciferine, Rutin, Hyperin, Demethylcocaluerine, Quercetin-3-O-β-D-glucuronide, (+)-1(R)-coclaurine, (−)-1(S)-norcoclaurine, Linalool, Luteolin glucoside, Dehydroanonaine, Anonaine, Armepavine, Asimilobine, Lirinidine, β-sitosterol, Liriodenine, Nornuciferine Quercetin, Dehydronuciferine, N Dehydroemerine, Isoquercitrin, methylcoclaurine, N-methylasimilobine, Roemerin, N-norarmepavine, N-methylisococlaurine, Kaempferol and derivatives	*Sahasrapatra, Neelkamal*	Nourishing, cardiotonic, calming, promotes strength and relieves bleeding disorders viz. epistaxis, hemoptysis, hematuria, and menorrhagia
*Oroxylum indicum* Vent.	Baicalein, biochanin a, 8, 8’ bis-baicalein, chrysin, ellagic acid, 6–hydroxy luteolin, oroxylin a, oroxoloside methyl ester, β-sitosterol, scuttellarien, ursolic acid, chrysin-7-ogentiobioside, baicalein-7-odiglucoside, scutellarein-7-o-glucopyranoside, aequinetin, chrysin-6-c-β-D-glucopyranosyl-8- c-α-l arabinopyranoside, pinocembrin, pinobanksin, lupeol, 2α-hydroxyl lupeol, echinulin, adenosine, and dimethyl sulfone	*Aralu, bhut-vriksha, Shyonak*	General weakness, hepatoprotective, antimicrobial, nephroprotective, cardiotonic
*Phaseolus trilobus*	α- pinene, carvone, pulgeone, dalbergioidin, kievitone, phaseollidin, flavonoid glycosides viz quercetin, kaempferol, vitexin, and isovitexin	*Mudagparni*	Tonic, aphrodisiac, mild sedative, antioxidant, promote strength, improves semen and sperm quantity
*Phyllanthus niruri* Linn.	Rutin, quercetin, gallocatechin, nirurin, nruriflavone, quercetol, astragalin, quercitrin, limonene, p-Cymene, lupeol, ellagic acid, gallic acid, elligitannin, hexahydroxyldiphenoyl moiety and methyl brevifolincarboxylate, phyllanthin, hypophyllanthin, niranthin, lintetralin, phyltetralin, nirtetralin, and isolintetralin	*Bhumyamalaki, Bhumi-awala*	Antioxidant, cholagogue, laxative, hepatoprotective, anticancer, antiviral
*Piper longum* Linn.	Carboxylic acids and derivatives, cinnamic acids and derivatives, isoflavonoids, napthalenes, oxanes, phenanthrenes and derivatives, phenol ethers, phenylpropanoic acids, pteridines and derivatives, pyridines and derivatives and steroid and its derivatives	*PippaIi*	Antitussive, stimulant, bronchodilator, tonic, bioavailability enhancer, carminative, relieves respiratory infections and hepatitis
*Pistacia integerrima* Stewart-ex Brandis	β-pinene, sabinene, α-pinene and limonene while terpinen-4-ol, α-terpinol, α-pinene, Pistacienoic acids, Hydroxydecanyl arachidate, Octadecan-9, 11-diol-7-one, β-Sitosterol, and Pisticialanstenoic acid	*Kakdasingi*	Bronchodilator, expectorant, carminative, antitussive, digestive, cholagogue
*Premna illtegrifolia* Linn.	p-methoxy cinnamic acid, linalool, linoleic acid, β-sitosterol, luteolin, iridoid glycoside, premnine, ganiarine, ganikarine, premnazole, aphelandrine, betulin, caryophellen, premnenol, premna spirodiene, and clerodendrin-A	*Arni, Agnimanth*	Laxative, antitussive, digestive
*Pterocarpus santalinus*	Santalin A, B, and Y, pterocarptriol, isopterocarpalone, pterocarpodiolones, β-eudesmol, cryptomeridiol, sesquiterpenes, β-sitosterol, lupeol, epicatechin, lignans, pterostilbenes, cinnamic acid, acetophenones, phenylacetic acid, lignans, coumarins, benzophenones, xanthones, 3-hydroxybenzoic acid, gentisic acid, α and β resorcylic acid, and vanillic acid	*Raktachandan*	Tonic, aphrodisiac, antipyretic, anti-hyperglycemic, diaphoretic anticancer, protective and antimicrobial effect on genitourinary and bronchial tract mucosa
*Sesamum indicum* Linn.	(+)-samin, (-)-asarinin, sesamol, (+)-sesamolin, (+)-sesamin, (+)-(7S,80R,8R)-acuminatolide, (-)-piperitol, and (+)-pinoresinol	*Tiltaila*, Sesame oil	Nutritive, demulcent, aphrodisiac, antioxidant, wound healing, radioprotective, anti-inflammatory
*Sida cordifolia* Linn.	Ephedrine, pseudoephedrine, sterculic, malvalic, coronaric acid, betaphenethylamine, hypaphorine, ecdysterone, indole alkaloids, palmitic, stearic and β–sitosterol	*Bala, Bariyara*	Cardiotonic, aphrodisiac, strength/vitality promoter
*Solanum indicum* Linn.	Indiosides A, B, C, D, F, Protodioscin, Carpesterol, Isoanguivine, Solanidine, Solasodine, Solamargine, Solavetivone, Solafuranone, Scopoletin, N-p-trans-Coumaroyltyramine, N-Trans-Feruloyltyramine, 7-Hydroxy-6,8-Dimethoxy-3-(40-Hydroxy-30-Methoxyphenyl)-Coumarin, Isofraxidin, Fraxetin, Indicumine A, B, C, D, F, Arteminorin, Cleosandrin, 4, 4’-biisofraxidin, β-Sitosterol, Daucosterol, Diosgenin, Lanosterol, Trilinolein, Oleodilinolin, and Palmitodilinolin	*Brihati, Barikateri, Vanbhanta*	Cardiotonic, astringent, carminative and digestive
*Solanum xanthocarpum* Schrad. and Wendi.	Lupeol, oleanolic acid, ursolic acid, β-sitosterol, campesterol, ergosterol, withanolide B	*Kantakaari, Chotikateri*	Mucolytic, expectorant, anti-allergic, bronchodilator and relieves flu
*Stereospermum suaveolens* Prodr.	Sterekunthal B, sterochenol A&B, lapachol, dehdro-a-lapachone, apigenin, scutellarein, sterolensin, dinatin, and dinatin-7-glucuroniside	*Paatla*	Tonic, digestive, cardiotonic, anti-inflammatory, blood purifier, antianemic
*Teramnus labialis* Spreng.	Vitexin, bergenin, daidzin and 3-O-methyl-D-chiro–inositol	*Mashaparni*	Improves vigor and virility, aphrodisiac, relieves debility and fatigue
*Terminalia chebula* Retz.	Chebulin, ellagic acid, 2,4-chebulyl-D-glucopyranose, arjunglucoside I, arjungenin, chebulinic acid, gallic acid, ethyl gallate, punicalagin, terflavin A, terchebin, luteolin and tannic acid	*Harad, Haritaki*	Neurotrophic, rejuvenative, carminative anthelmintic, nervine tonic, appetite stimulant
*Tinospora cordifolia*	Tinosporine, tinosporide, tinosporaside, cordifolide, cordifol, heptacosanol, clerodane furano diterpene, diterpenoid furano lactone, tinosporidine, columbin, b-sitosterol, Berberine, palmatine, tembertarine, magniflorine, choline and tinosporin	*Guduchi, Chinnodbhava*	General tonic, immunomodulator, cytoprotective, genoprotective, adaptogenic
*Tribulus terrestris* Linn.	Tigogenin, neotigogenin, gitogenin, neogitogenin, hecogenin, neohecogenin, diosgenin, chlorogenin, ruscogenin, sarsasapogenin, protodioscin protogracillin, kaempferol, kaempferol-3-glucoside, kaempferol-3-rutinoside, tribuloside, quercetin 3-*O*-glycoside, quercetin 3-*O*-rutinoside, and kaempferol 3-*O*-glycoside	*Gokhru, Gokshur*	Aphrodisiac, mood elevator, diuretic and cardiotonic
*Uraria picta* Desv.	5, 7-dihydroxy-2’-methoxy-3’, 4’- methylenedioxyisofla-vanone, and 4’, 5’-dihydroxy-2’, 3’-dimethoxy-7-(5-hydroxychromen-7yl)-isoflavan-one, isoflavanones, triterpenes and steroids	*Prishnaparni, Pithawan*	General weakness, nervine tonic, cardiovascular disorders
*Vitis vinifera* Linn.	Oleanolic and betulinic acids, stilbenoid, daucosterol, E-resveratrol, E-ε-viniferin, (-)-epicatechin, catechin, gallocatechin, 6′-O-acyldaucosterols, 1,2-di-O-acyl-3-O-β-D-galactopyranosyl glycerols, gallic acid, p-coumaric, caffeic and ferulic acids, anthocyanidin-3-O-glucosides, malvidin-3-O-glucoside, peonidin-3-O-glucoside and cyanidin-3-O-glucoside	*Draaksha, Munnakka*	Nutritive, aphrodisiac, cardiotonic, diuretic, demulcent, laxative, hepatoprotective, cures thirst and asthma
Indian *Cow Ghee*	Monounsaturated fats, conjugated linoleic acid, antioxidants, vitamins A, E, D, K, and beta carotene	*Go-ghrita*	Nutritive, antioxidant, strengthens the immunity, anticancer, improves overall physical and mental strength
Natural honey (derived from honey bees)	Phenolics (e.g., gallic, syringic, benzoic, transcinnamic, protocatechuic, p-coumaric, caffeic acids), Flavonoids (e.g., catechin, kaempferol, naringenin, luteolin, pinostrobin, apigenin), fructose, oligosaccharides (palatinose, isomaltose and alpha-cyclodextrin), carotenoids, cholines, kynurenic acid, enzymes (glucose oxidase, diastase, invertase, phosphatase, catalase peroxidase), vitamins (B1, B2, B3, B5, B6, B9, C, phyllochinon), minerals/trace elements (Na, Ca, K, Fe, Mg, P, Zn, Cu, Cr, Mn, S, B, Se, Mo, Co, F, I, Si)	*Madhu*	Anti-infective, immunomodulator, wound healing, antioxidant, antiaging, relieves cough and cold, antiseptic, sore throat, antiulcer
Sugar candy	Disaccharides (sucrose)	*Sharkara*	Sweetener, provides calories/energy
*Asparagus racemosus* Willd	Shatvarin I to VI, oligospirostanoside, aspargamine A, dihydrophenantherene, racemofuran, quercitin, rutin, hyperoside, sitosterol, 4, 6-dihydryxy-2-O (-2-hydroxy isobutyl) benzaldehyde, undecanyl cetanoate, sarsapogenin, γ-linoleinic acids, diosgenin, and quercetin 3-glucourbnides	*Shatavari, Shatavar*	Aphrodisiac, nutritive, galactogogue, tonic, antiulcer, antioxidant, good for eyes
*Dioscorea bulbifera* Linn.	Dioscoreanoside A-K, diosbulbisin A-D, diosbulbisides A-C, diosgenin, sinodiosgenin, diosbulbin A-P, 8-epidiosbulbin E acetate, Bafoudiosbulbin A-G, quercetin-3-O-β-dglucopyranoside, neoxanthin, β-sitosterol, catechin, vanillic acid, isovanillic acid and glycoside derivatives	*Varahikand, Varahi*	Aphrodisiac, antiulcer, tonic, promotes vigor and strength
*Ipomoea digitata* Linn.	Taraxerol, taraxerol acetate, N-butyl-β-Dfructopyranoside, octadecyl (E)-p-coumarate, umbelliferone, scopoletin, scopolin, scoparone, scopoletin and taraxerol	*Vidaarikand*	Aphrodisiac, antioxidant, galactogogue, nervine tonic, relieves debility and spermatorrhea
*Withania somnifera* Dunal	Cuscohygrine, anahygrine, tropine, pseudotropine, anaferine, isopelletierine, withananine, withananinine, pseudo-withanine, somnine, somniferine, somniferinine, withanine, withasomnine, visamine, chlorogenic acid, and withaferin A	*Ashwagandha, Asgandh*	Aphrodisiac, adaptogenic, antioxidant, cytoprotective, neuroprotective, nootropic, antistress, promotes strength
**Special Additives**			
*Abhraka Bhasma*	Ayurvedic nanosized mineral preparation of incinerated biotite mica, Nature of compound: multimineral cocktail	*Abhraka Bhasma*	General debility, aphrodisiac. cardiotonic, cellular regenerator, useful in digestive impairment, malabsorption syndrome, asthma and cough
*Shukti Bhasma*	Ayurvedic nanosized mineral preparation of calcined Pearl oyster, Nature of compound: Calcium Carbonate	*Shukti Bhasma*	Antacid, antiarrhythmic, antihistaminic calcium supplement, neurotrophic, cardiotonic and promotes bone strength
*Shringa Bhasma*	Ayurvedic nanosized preparation of calcined Deer horn, Nature of compound: Calcium Phosphate	*Shringa Bhasma*	Expectorant, effective in pleurisy, pneumonia, tuberculosis, productive cough
*Siddha Makardhawaja*	Ayurvedic metallo-mineral preparation having purified and processed gold, mercury and sulphur in 1:8:24 ratio, Nature of compound: HgS (with nanotraces of gold)	*Makardhawaja*	Antiaging, aphrodisiac, cardiovascular tonic, help to cure male impotency, erectile dysfunction, premature ejaculation
*Eugenia caryophyllus* Linn.	Acetyl eugenol, β-caryophyllene, vanillin, crategolic acid, bicornin, methyl salicylate, gallotannic acid, seugenin, rhamnetin, kaempferol, eugenitin, triterpenoids, oleanolic acid, campesterol, stigmasterol, sesquiterpenes, etc.	*Loung,* Clove	Antiseptic, antimicrobial, aromatic, stimulant and anti-inflammatory

**Table 2 biomolecules-09-00161-t002:** Eight “missing” ingredients of CP—the *Ashtavarga*.

Botanical Name	Folk Name	Family	Substitutes Used in Commercial CP
*Crepidium acuminatum*	*Jeevak*	Orchidaceae	*Vidaarikand* (*Ipomoea digitata*)
*Malaxis muscifera*	*Rishbhak*	Orchidaceae	
*Habenaria intermedia*	*Riddhi*	Orchidaceae	*Varahikand (Dioscorea bulbifera)*
*Habenaria edgeworthii*	*Vriddhi*	Orchidaceae	
*Roscoea purpurea*	*Kakoli*	Zingiberaceae	*Ashwagandha* (*Withania somnifera*)
*Lilium polyphyllum*	*Kshirkakoli*	Liliaceae	
*Polygonatum verticillatum*	*Meda*	Liliaceae	*Shatavari* (*Asparagus racemosus*)
*Polygonatum cerhifolium*	*Mahameda*	Liliaceae	

**Table 3 biomolecules-09-00161-t003:** Comparative test performance scores of leading market brands of Chyawanprash.

	Brand	Weight %	Baidyanath Kesari Kalp	Apollo Pharmacy	Dabur	Humdard	Divya (Patanjali Yogpeeth)	Himalaya	Himani (Sona Chandi)	Zandu
Parameter										
**Packing size (kg)**			01	01	01	01	01	01	01	01
**Maximum Retail Price (₹)**			520	220	210	210	200	220	230	185
**Physicochemical Analysis**										
Antioxidant		08	6.94	7.77	6.88	7.71	6.97	7.51	6.09	7.74
Phenolic Compound		05	4.02	4.67	3.99	4.5	4.05	4.42	3.78	4.49
Vitamin C		08	6.22	08	07	7.2	6.27	5.98	6.88	4.54
Piperine		04	3.48	3.4	3.56	04	3.44	3.72	2.8	3.36
Steroids		04	04	04	04	04	04	04	04	04
Fat		04	3.74	3.7	3.15	3.67	3.51	3.63	2.8	3.8
Protein		03	03	2.1	2.27	2.18	2.4	2.27	2.3	2.13
Carbohydrate		04	3.94	3.41	3.25	3.04	3.6	3.55	3.01	3.59
Calorific Value		03	2.55	2.79	2.61	03	2.6	2.66	2.58	2.73
Crude Fiber		02	1.88	1.4	1.89	1.98	1.69	02	1.93	1.59
Dietary Fiber		02	1.83	1.75	1.98	02	1.8	1.73	1.71	1.77
Pesticides		04	04	04	04	04	04	04	04	04
Heavy Metals		06	06	06	06	06	06	06	06	06
Aflatoxin		06	06	06	06	06	06	06	06	06
Sugar		03	2.83	2.46	2.82	2.85	2.72	2.45	2.84	2.4
pH		03	2.47	2.18	2.29	2.1	2.85	2.49	2.4	2.37
Water Content		03	2.41	2.66	2.77	2.97	2.56	2.5	2.82	2.68
Total Ash		02	1.77	02	1.44	1.96	1.69	1.76	1.58	1.4
Acid Insoluble Ash		02	1.98	1.92	1.97	1.76	02	1.96	1.4	1.94
Microbiological assay ^α^		06	06	06	06	06	06	06	06	06
Organoleptic analysis ^β^		13	10.87	9.35	11.05	8.16	9.96	9.02	8.99	6.98
**General Parameters**										
Packing		02	02	02	02	02	02	02	02	02
Marking		03	2.7	03	03	2.7	2.7	2.4	03	2.7
Overall Score		100	90.63	90.56	89.92	89.78	88.81	88.05	84.91	84.21

^α^ Total plate count, yeast and mold count and pathogens; ^β^ Color, appearance, odor/flavor, taste, aftertaste sensation; rating: >90—Excellent, 71–90—Very Good, 51–70—Good, 31–50—Average, up to 30—Poor.

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
