# Peer review of "Chyawanprash: A Traditional Indian Bioactive Health Supplement"

_biomolecules, 2019, doi:10.3390/biom9050161_

Round 1
Reviewer 1 Report
Interesting review.
The English is good grammatically speaking, but needs some scientifically like-like formulations. Propositions like “Former signify the name of a sage” are not acceptable.
Same goes with some info that needs better referencing, although the author did an extensive bibliographical research for this meritoriu9os review paper.
If these aforementioned matters are resolved the scientific efforts of the authors can be much more easily evidenced.
Also, I would have expected more on the antioxidant aspects and related vitamic C deficiency, which is still interesting described by the authors.
Author Response
Point 1: Interesting review.
The English is good grammatically speaking, but needs some scientifically like-like formulations. Propositions like “Former signify the name of a sage” are not acceptable.
Response 1: Thank you for your careful reading and valuable suggestions.
Point 2: Same goes with some info that needs better referencing, although the author did an extensive bibliographical research for this meritoriu9os review paper.
If these aforementioned matters are resolved the scientific efforts of the authors can be much more easily evidenced.
Response 2: We are grateful for your time and constructive comments on our manuscript.
We have incorporated the points suggested by you and worked on better referencing. More scientific evidence based information is now added to make the paper more significant and interesting for the broad readership of Biomolecules Journal. More info is incorporated viz. major active biomolecules, more inputs in pharmaceutical process, quality specifications of CP, and supportive pharmacological and clinical reports in antioxidant aspects and health benefits of CP etc. Total references are now 157 (earlier: 102).
Point 3: Also, I would have expected more on the antioxidant aspects and related vitamic C deficiency, which is still interesting described by the authors.
Response 3: Thank you for pointing this out. We have incorporated all the relevant information as per your suggestion.
All the Thanks for careful reading again.
Your inputs would surely improve the quality of our article.
Reviewer 2 Report
The article by Rohit Sharma et al. presents an extensive description of a complex ayurvedic traditional medicinal product by addressing very brad topics including composition, preparation, health related indications and commercial aspects. Unfortunately the level of detail provided however is quite superficial in that not enough details of scientific interests are provided regarding the main active molecules, their mechanisms of action and scientific based evidences for the numerous beneficial effect described is also missing, as the author also clearly declare. Moreover the authors also describe that commercial products do not follow a standard operating procedure and that the product is available on the market in several different preparation. Altogether these elements further weaken the scientific solidity of the work that is in general quite superficial.
Author Response
Point 1: The article by Rohit Sharma et al. presents an extensive description of a complex ayurvedic traditional medicinal product by addressing very brad topics including composition, preparation, health related indications and commercial aspects.
Unfortunately, the level of detail provided however is quite superficial in that not enough details of scientific interests are provided regarding the main active molecules, their mechanisms of action and scientific based evidences for the numerous beneficial effects described is also missing, as the author also clearly declare.
Response 1:
Thank you for pointing out the faults. We are grateful for your careful reading and constructive comments on our manuscript.
We have incorporated the points suggested by you and thoroughly revised the paper to provide in depth scientific information related to our topic. More scientific evidence based information is now added to make the paper more significant and interesting for the broad readership of Biomolecules Journal. More info is incorporated viz. major active biomolecules (new column added in Table 1), more inputs in pharmaceutical process, quality specifications of CP, and all available pharmacological and clinical reports to support the several claimed health benefits of CP etc. Total references are now 157 (earlier: 102).
Point 2: Moreover, the authors also describe that commercial products do not follow a standard operating procedure and that the product is available on the market in several different preparation. Altogether these elements further weaken the scientific solidity of the work that is in general quite superficial.
Response 2:
CP is a widely used supplement in Indian subcontinent having the market value over a billion USD and is witnessing a upsurge in popularity overseas too. So the paper detailed the ideal classical multistep procedure for preparation of CP and highlighted regarding Ayurvedic Pharmacopoeia of India (API), that has published a monograph on CP having a brief info on method of preparation and quality specifications. On the contrary, individual firms are manufacturing such a preparation have their in-house specifications for quality of CP, which are not in the public domain. Non-compliance with ancient manufacturing guidelines and deviation from original recipe is a common malpractice in pharma sector. The original formula is wiped out in between the practices of preparation and immoral marketing tactics to make it more presentable and appealing than competitors’ product. It will no longer be called CP if there is any change in ingredients of the traditional formula.
So we tried to encompass extensively all the available info on this traditional Indian nutraceutical to update the scientific fraternity apropos its role in public health.
All the Thanks for your careful reading and valuable suggestions.
Your inputs would surely improve the quality of our article.
Reviewer 3 Report
The work is well written and very interesting, especially for people from other cultures. In my opinion it is suitable for publication.
Author Response
Point 1: The work is well written and very interesting, especially for people from other cultures. In my opinion it is suitable for publication.
Response 1:
Thank you for your careful reading and valuable comments.
We are grateful for your time and positive comments on our manuscript.
Round 2
Reviewer 2 Report
The manuscript have been significantly improved by discussing and referencing appropriate clinical data. In summary the review supports its publication considering that the arguments have been accurately described.
Nonetheless, I have some general concerns regarding the scientific meaning of the overall product and application of scientific discussion of its mechanism of action taking into consideration the complex nature of the product itself comprising dozens of active compounds with very broad biological effects on different targets. Strictly speaking in such a complex mixture it is very difficult to describe in a detailed manner efficacy supported by mechanism of actions. Identifying biochemical and molecular targets would significantly elevate the overall meaning of the review.
Moreover, in the present form, synergistic or antagonistic effects of compounds are difficult to deal with and are not evincible from the present literature.
In conclusion I believe that the authors have done a good work in collecting information on the composition, preparation and biological effects of the formula. The scientific evidences describing the mechanism of action of the single compound in such a complex formula, further complicated by various protocol of preparation weakens the scientific credibility of the product.
I believe that the authors should at least highlight in their conclusion that, despite the traditional implementation in ayurvedic medicine and the reported efficacy evidences, there is a requirement of controlled experiments on the effect of the main active compounds and their synergistic or antagonistic effect in order to clarify their mechanism of action. This could lead also to improvement of the traditional preparation that is not necessarily the optimal version. In fact strict compliance with the centuries old recipe is not per se a guarantee of success in absence of appropriate scientific evidences
Author Response
Point 1: The manuscript have been significantly improved by discussing and referencing appropriate clinical data. In summary the review supports its publication considering that the arguments have been accurately described.
Response 1:
We are grateful for your careful reading and positive comments on our revised manuscript.
Point 2: Nonetheless, I have some general concerns regarding the scientific meaning of the overall product and application of scientific discussion of its mechanism of action taking into consideration the complex nature of the product itself comprising dozens of active compounds with very broad biological effects on different targets. Strictly speaking in such a complex mixture it is very difficult to describe in a detailed manner efficacy supported by mechanism of actions. Identifying biochemical and molecular targets would significantly elevate the overall meaning of the review.
Moreover, in the present form, synergistic or antagonistic effects of compounds are difficult to deal with and are not evincible from the present literature.
In conclusion I believe that the authors have done a good work in collecting information on the composition, preparation and biological effects of the formula. The scientific evidences describing the mechanism of action of the single compound in such a complex formula, further complicated by various protocol of preparation weakens the scientific credibility of the product.
I believe that the authors should at least highlight in their conclusion that, despite the traditional implementation in ayurvedic medicine and the reported efficacy evidences, there is a requirement of controlled experiments on the effect of the main active compounds and their synergistic or antagonistic effect in order to clarify their mechanism of action. This could lead also to improvement of the traditional preparation that is not necessarily the optimal version. In fact strict compliance with the centuries old recipe is not per se a guarantee of success in absence of appropriate scientific evidences
Response 2:
All the Thanks for your suggestions and constructive comments.
We have incorporated the points suggested by you in our manuscript.